# Identifying research priorities for road safety in Nepal: a Delphi study

Puspa Raj Pant ,[1] Pranita Rana,[2] Kriti Pradhan,[2] Sunil Kumar Joshi ,[3] Julie Mytton [1]

[1]Faculty of Health and Applied Sciences, University of the West of England, Bristol, UK
[2]Nepal Injury Research Centre, Kathmandu Medical College, Kathmandu, Nepal
[3]Community Medicine, Kathmandu Medical College and Teaching Hospital, Kathmandu, Bagmati, Nepal

**Correspondence to**
Dr Puspa Raj Pant;
pant.puspa@gmail.com

## ABSTRACT

**Objective** To identify and prioritise the research needed to help Nepali agencies develop an improved road safety system.

**Design** Delphi study.

**Setting** Nepal.

**Participants** Stakeholders from government institutions, academia, engineering, healthcare and civil society were interviewed to identify knowledge gaps and research questions. Participants then completed two rounds of ranking and a workshop.

**Results** A total of 93 participants took part in interviews and two rounds of ranking. Participants were grouped with others sharing expertise relating to each of the five WHO 'pillars' of road safety: (1) road safety management; (2) safer roads; (3) safer vehicles; (4) safer road users and (5) effective postcrash response. Interviews yielded 1019 research suggestions across the five pillars. Two rounds of ranking within expert groups yielded consensus on the important questions for each pillar. A workshop involving all participants then led to the selection of 6 questions considered the most urgent: (1) How can implementing agencies be made more accountable? (2) How should different types of roads, and roads in different geographical locations, be designed to make them safer for all road users? (3) What vehicle fitness factors lead to road traffic crashes? (4) How can the driver licensing system be improved to ensure safer drivers? (5) What factors lead to public vehicle crashes and how can they be addressed? and (6) What factors affect emergency response services getting to the patient and then getting them to the right hospital in the best possible time?

**Conclusions** The application of the Delphi approach is useful to enable participants representing a range of institutions and expertise to contribute to the identification of road safety research priorities. Outcomes from this study provide Nepali researchers with a greater understanding of the necessary focus for future road safety research.

## INTRODUCTION

Globally road traffic injuries (RTIs) are increasing, with an estimated 1.35 million deaths and up to 50 million non-fatal injuries in 2016.[1] Despite having only 1% of the world's vehicles, low-income countries have 13% of fatal RTIs. RTIs are the leading cause of death for children and young adults between 5 and 29 years globally and are an important cause of disability and poverty. RTIs have been estimated to generate losses of up to 6.5% of a low-income country's gross domestic product.[2]

The WHO World Report on Road Traffic Injury Prevention,[3] subsequent Road Safety Status Reports[1] and the WHO Save LIVES technical package of 22 evidence-based interventions[4] argue for a 'safe systems approach'[5] to reduce road dangers and the numbers of people killed and seriously injured on the roads. This approach recognises the essential contribution of different sectors to create a system that keeps road users safe. The WHO published the Global Plan of Action for Road Safety 2011–2020[6] alongside the United Nations and this plan of action recommended five 'pillars'; road safety management (pillar 1), safer roads and mobility (pillar 2), safer vehicles (pillar 3), safer road users (pillar 4) and postcrash response (pillar 5). Action across all five pillars can contribute to reduced RTIs. Nepal has been a cosponsor of these principles, but progress has been limited.

A large road construction programme in Nepal has seen over 15 000 km of new blacktop, gravel and earthen roads built by federal, provincial and local governments in the last 5 years[7] and there are plans to have

a total of 13 500 km blacktopped road by 2023/2024.[8] Many new roads do not have proven safety features and are poorly maintained. The roads in the hills are considered to be dangerous because of landslides in addition to frequent road crashes due to poor engineering or poor safety infrastructure.[9] The Department of Transport Management in the Government of Nepal produces vehicle registration statistics that show more than half (53%) of the 3.22 million motorised vehicles in Nepal were registered between July 2013 and July 2018 and about 78% of total registered vehicles were motorcycles.[10]

## Strengths and limitations of this study

► Ninety-three Nepali experts (70% of 133 approached) participated, bringing perspectives from road construction, vehicle management, transport management and postcrash response.
► Most participants had a remit for national road safety, however, 83/93 (89%) were from organisations based in Kathmandu valley, which may have risked a focus on urban and highway crashes.
► The research questions identified were ranked by the participants individually as well as discussed during group meetings to achieve consensus.
► We were able to retain a high proportion of participants through the study: 64/93 participants took part in round 3 (69% retention).

Nepal lacks a funded road safety implementation plan, a national ambulance service, or globally recognised vehicle standards. The national helmet-wearing law is not enforced for motorcycle passengers and there is no legislation for passenger seatbelt use, child restraints, or mobile phone use while driving. Data are limited and of poor quality; WHO estimates of road traffic fatalities in Nepal in 2016 (4622) are more than double those recorded by the Traffic Police (2006), and there are no routinely published estimates of deaths by road user category available.[1] Nepal's Health Management Information System recorded over 100 000 hospital visits for the treatment of orthopaedic problems secondary to road traffic events in the year 2017/2018 indicating the significant burden of RTIs on health systems.[11] Road traffic crashes and injuries in Nepal are rising despite existing legislation.[12 13] Tackling RTIs was a priority in the government's Health Sector Strategy 2015–2020.[14] A National Road Safety Action Plan 2013–2020[15] was acknowledged but not ratified by Parliament. Neither document specified the research required to support the delivery of improved road safety.

To improve road safety, coordinated efforts are needed across the road transport system. Research is vital to optimise decision-making. Current initiatives in Nepal for the control and prevention of road traffic crashes and their consequences are not based on local evidence. Therefore, this study aimed to involve a wide range of experts and participants representing stakeholder organisations to identify the research needed to help agencies in Nepal develop a safe systems approach to road safety, and achieve a consensus about which studies should be prioritised.

## METHODS

This study used the Delphi approach[16–18] to develop a consensus on a prioritised list of road safety research questions. Five groups of stakeholders in Nepal were engaged. The roles and experience of participants were relevant to each of the five WHO pillars of road safety. The study was conducted in two stages: first, interviews were conducted with stakeholders to identify a range of possible research questions, and second, participants completed two rounds of ranking the research questions in order of importance. Each of the five road safety pillars was studied separately. Five interview topic guides were developed in the Nepali language, based on the activities

recommended for each of the five WHO pillars of road safety (online supplemental file 1).

### Participant recruitment

Potential study participants were identified through existing networks and multisector stakeholder groups on road safety and first response convened by the Nepal Injury Research Centre. Networks included third sector and advocacy organisations for road safety. Participants helped identify further potential participants through a snowballing approach where they advised the research team of individuals who may be appropriate to invite to take part. We aimed to recruit 20–25 participants for each of the five pillars. Potential participants were contacted by telephone and were provided with information about the study and their interest in our research was confirmed. For participants expressing an interest, written information regarding the study and a consent form were sent to the potential participants via email. All the recruitment took place during the novel COVID-19 pandemic and therefore most of the interviews were completed remotely, by phone or video call. For these participants, consent was recorded verbally at the start of the interview or was collected before participation via email. Later in the pandemic, it became feasible to engage some participants face to face. For these participants consent was collected at this meeting.

### Data collection and analysis

In round 1, we conducted interviews with participants in which we asked what additional data or information would help them in their job and reduce RTIs. We explored the barriers they faced when tackling road safety. Most of the interviews were conducted using online platforms such as MS Teams, Zoom, Google Meet or Viber, and some interviews were conducted over the telephone. Towards the end of the data collection period, and when COVID-19 pandemic restrictions allowed, we conducted a small number of face-to-face interviews where this was the preference of the participants. In these circumstances, mitigations against infection, such as social distancing and the wearing of face masks, helped protect both participants and researchers. Interviews were conducted in the Nepali language and audiorecorded. Audiorecordings were listened to several times. Information relating to perceived gaps in research or evidence was documented as potential research questions on a spreadsheet, in English. For each group of stakeholders, approximately 200 research suggestions were generated from the interviews. Many of the participants raised similar issues, therefore it was possible to cluster the questions into groups, and to formulate a single question to represent that area of research need. The grouping stage was completed collaboratively by the whole research team to ensure that questions were treated equally and the process consistently applied. A reduced list of about 30 questions was achieved, identifying the research and evidence needs relating to each pillar of road safety.

For round 2, the research questions from the reduced list were uploaded to an online survey tool (Qualtrics) in both English and Nepali languages. The link to the survey was distributed to the participants via email or Viber message. Participants were asked to give their opinion on the importance of each research question using a 5-point Likert scale: not Important, slightly important, moderately important, important and very important. Reminders to complete the survey were sent via email and individual phone calls after 1 week and followed up again 2–3 days later. Completed surveys were exported from Qualtrics and analysed in MS Excel. Survey results were collated to identify the number of participants who rated each question as 'important' or 'very important'. Questions where a significant majority of participants had scored them 'important' or 'very important' were retained as prioritised questions. For pillars 1, 3, 4 and 5, we retained questions where ≥70% of the participants rated the questions as 'important' or 'very important'. For pillar 2, we retained questions where ≥80% of participants rated at these levels, since a greater proportion of the questions were considered important. We used these threshold values based on published Delphi studies.[19 20]

For round 3, participants were invited to a real-time online workshop where the prioritised questions were presented and discussed. The workshop was designed to allow the participants to share their views and listen to each other's opinions regarding which issues were the most important to research. These workshops were recorded and shared with those who were not able to join. Following the workshop, a Qualtrics survey was sent to all participants again, this time listing only those questions prioritised from round 2. Participants were again asked to score each question as either not important, slightly important, moderately important, important or very important. Reminders were sent to the participants after 1 week and followed up again after 2–3 days. Completed surveys were exported to MS Excel and collated to identify the number of participants considering each question 'important' or 'very important'. This resulted in the final prioritised list of research questions for each pillar of road safety.

The research team completed rounds 1, 2 and 3 for one pillar before moving on to the next pillar. The interviews started on 12 July 2020 and were completed on 14 February 2021. Due to the COVID-19 pandemic, where government officials and clinical staff were not easily available to participate, stakeholders in pillars 1 and 5 were left until later in the study when the peak of the first wave of COVID-19 in Nepal had passed.

### Overarching consensus workshop

A final online consensus workshop was organised where the top-ranked research questions from all five pillars were shared with all the participants, stakeholders from our advisory groups and invited key decision-makers. A facilitated discussion explored the understanding of what the different research options could provide and how

that new evidence could potentially be used. Using online voting software (Mentimeter, https://www.menti.com), participants were encouraged to vote for one research question from each pillar that they considered needed to be addressed the most urgently. The questions considered most urgent were presented back to the group.

### Patient and public involvement

Through community engagement and involvement, we engaged individuals with diverse views on road safety, ranging from road users to those with decision-making authority for road development, management and traffic regulation.

## RESULTS
### Study participants

Out of a total of 133 potential participants identified and contacted, 93 individuals were recruited and took part in interviews covering all five road safety pillars. Two participants had expertise relevant to more than one pillar, and therefore, took part in two interviews; one for each pillar. Participants were from a range of organisational and professional backgrounds, including government institutions, academia, road safety engineers, clinicians, civil society organisations and all had an interest or remit that addressed one or more of the five pillars of road safety. Some of the experts in our list, when contacted, suggested the name of other stakeholders. Out of 93 participants, 83 were from Kathmandu valley and represented organisations with the remit to work or influence road safety nationally. Ten participants were from outside Kathmandu and added value to the study by providing local contexts. The participants' background characteristics are summarised in table 1.

Across all five pillars, we identified a total of 1019 research suggestions from the 95 interviews completed in round 1. Collating similar questions reduced this to 141 questions across the five pillars. Seventy-six (80%) participants took part in round 2, through which the list of questions was reduced to 91 questions. Forty (43%) participants took part in an online workshop before further ranking in round 3 which was completed by 64 (69%) participants and resulted in a total of 30 prioritised questions. Figure 1 shows the stages of the Delphi study and the number of participants in each round. Attrition of participants was greatest for the group discussing Pillar 1 (road safety management), where 10/21 (48% participants) dropped out between round 1 and round 3. Attrition was least in the group discussing pillar 2 (safer roads) where only 3/18 (17%) of participants were lost.

The high attrition of participants in pillar 1 was not unexpected since many of these participants worked in government positions and it was difficult for them to prioritise attendance during the COVID-19 pandemic. Figure 2 illustrates participant attrition throughout the study.

**Table 1** Organisational/professional background of the participants

| Organisational/professional background | Total | Male | Female |
|---|---|---|---|
| Government organisation (secretaries, govt officers, police, political representatives) | 33 | 30 | 3 |
| Clinician, nurse, physiotherapist | 10 | 8 | 2 |
| Road safety engineer | 9 | 9 | 0 |
| Road safety advocacy | 8 | 5 | 3 |
| Academics | 7 | 6 | 1 |
| First aid/emergency/ambulance provider | 6 | 6 | 0 |
| Engineers' association | 4 | 4 | 0 |
| Transport worker | 4 | 4 | 0 |
| Automobile dealer | 3 | 3 | 0 |
| Federation of transport | 2 | 2 | 0 |
| Schools' organisation | 2 | 2 | 0 |
| Sustainable transport | 2 | 2 | 0 |
| Others (journalist and city planners) | 3 | 3 | 0 |
| **Total** | **93** | **84** | **9** |

Table 2 describes the number of research questions prioritised in each round, split by the pillars of road safety. The retention rate in this study was equivalent to that in other published Delphi studies[21] despite the COVID-19 pandemic.

The top-ranked research questions for the five pillars of road safety are presented in table 3. The research questions that were considered the most important cover a wide range of issues, including how to make existing processes more effective, how to assess the training needs of the road safety workforce, understanding the challenges of implementing existing road safety legislation, how to improve accountability for road safety, how to generate and disseminate better information to inform decisions and how to generate evidence that supports the economic argument for road safety.

A total of 56 people (47 participants and 9 key decision-makers) attended the workshop conducted at the end of the study where the list of the top-ranked research questions for each of the five pillars were presented. Using electronic voting software to identify the question within each Pillar considered to be the most urgent, 6 questions were prioritised. Two questions in pillar 4 were scored equally (table 4).

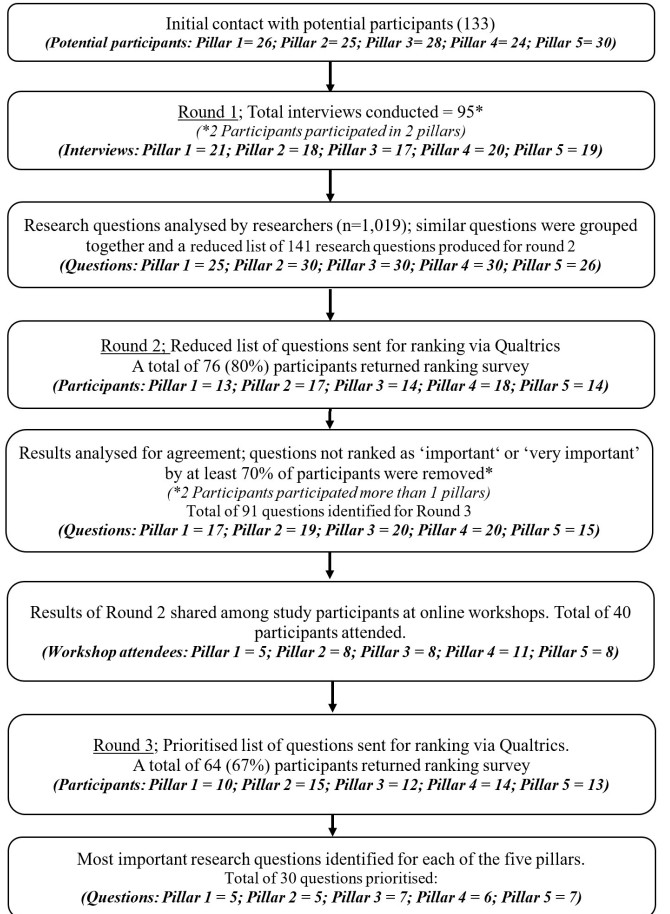

**Figure 1** Flow chart of the Delphi process.

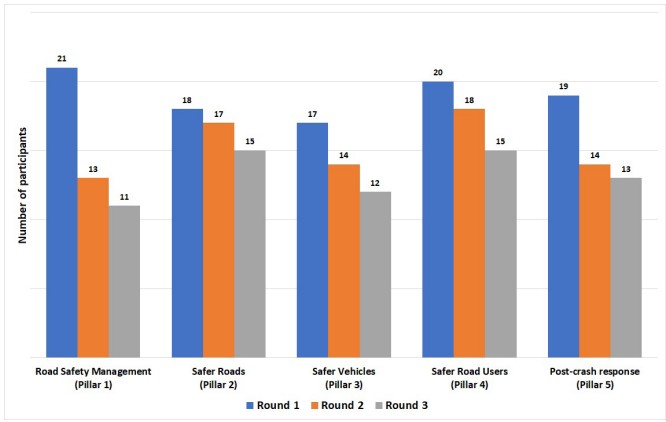

**Figure 2** Study participants retained in subsequent Delphi rounds, by pillar.

**Table 2** Research questions prioritised at each round, by pillar

| Pillar of road safety | Round 1 | | | Round 2 | | Round 3 |
|---|---|---|---|---|---|---|
| | Interview dates | No of interviews (online or by phone) | Research questions generated ('long list') | Grouped research questions ('reduced list') | No of 'important' or 'very important' research questions (above 70% consensus) | No of questions considered most important (Top five ranks) |
| Pillar 1 (road safety management) | 23 November to 22 January 2021 | 21 (21) | 183 | 25 | 17 | 5 |
| Pillar 2 (safer roads and mobility) | 13 July to 12 August 2020 | 18 (4) | 211 | 30 | 19* | 5 |
| Pillar 3 (safer vehicles) | 16 August to 15 September 2020 | 17 (17) | 217 | 30 | 20 | 7 |
| Pillar 4 (safer road users) | 23 September to 19 October 2020 | 20 (20) | 178 | 30 | 20 | 6 |
| Pillar 5 (postcrash response) | 05 January to 14 February 2021 | 19 (13) | 230 | 26 | 15 | 7 |
| Total | | 95 (75) | 1019 | 141 | 91 | 30 |

*80% consensus.

## DISCUSSION

This study is the first to our knowledge that has engaged such a wide group of participants to identify the research priorities relevant to the improvement of road safety in Nepal. The research team identified and invited 133 potential participants to join the study, and 70% (n=93) agreed to take part. Respondents included stakeholders from a range of organisational and professional backgrounds as well as geographical areas and included; officials in government institutions (ministerial secretaries, government officers, police, political representatives), clinicians, nurses, physiotherapists, engineers, academics, first responders, transport workers, automobile dealers, road users, members of the media and city planners. The proportion of women working in roles related to road safety in Nepal is low, and we were pleased to have been able to recruit 9/93 (10%) female participants, which is in line with official data on the Nepali workforce. The number of participants that should take part in a Delphi study is not prescribed and it can be anywhere above 10 persons; the number is guided by the scope of the problem and existing resources.[19 20] Overall, the retention of the participants until the third round of ranking was excellent however, rates varied between different pillars. The overall retention rate of 69% and 50% attendance at the final consensus workshop indicated the high level of interest in road safety research in Nepal. This response rate is higher than that reported by Marchau and van der Heijden[22] in a multicountry road safety study. Marchau and Van der Heijden[22] applied the Delphi technique to explore the policy aspects of implementing driver support systems. The authors used a questionnaire with specified answer options sent to international experts from the USA, Japan and Europe. In this study, 56% (65 out of 117) of invitees responded in the first round while only 40 responded in the third round.

Road safety research is a neglected issue in low-income and middle-income countries[23] and a lack of research capacity may be one reason for the limited progress to date. In Nepal, a policy review identified that institutional arrangements and resource allocation for road safety were inadequate.[13] The lack of coordination of road safety sectors is a challenge globally[24 25] as well as in Nepal. Many of the participants in this study had the opportunity to meet and discuss road safety with those working in other sectors, for the first time.

Other studies exploring aspects of road safety through the use of the Delphi technique have mostly come from high-income countries, except a few, such as Vietnam and Uganda. Studies have explored specific risk factors such as cell phone use and sleep deprivation in the USA,[26 27] and public bus safety in Italy.[28] Some studies focused on the need to improve postcrash care such as; strengthening trauma management in Vietnam,[29] prehospital emergency care in Iran,[30] postrecovery rehabilitation in Australia[31] and emergency medical services capacity in Uganda.[32] In Iran, Delphi studies have been conducted to inform the development of minimum datasets to study

**Table 3** List of top questions for pillars 1–5 with scores in rounds 2 and 3

| | Scores* | |
|---|---|---|
| **Pillar 1: Road safety management** | **R2** | **R3** |
| How can implementing agencies be made more accountable for road safety in urban and rural areas? | 92 | 91 |
| What are the barriers to conducting road safety audits at all stages of road construction and implementation of their recommendations? | 85 | 91 |
| How can urban and rural roads construction and management be governed to ensure improved road safety? | 92 | 91 |
| How can the traffic management system be improved to ensure it improves the safety of all road users? | 85 | 91 |
| What are the barriers to the implementation of existing laws related to road safety in Nepal? | 69 | 91 |
| **Pillar 2: Safer roads** | | |
| What is the effectiveness of different safety features installed on roads in terms of crash reduction? | 94 | 100 |
| What are the barriers and facilitators for achieving safer roads in Nepal? | 88 | 100 |
| What kind of institutional setup is needed at central, provincial, and local levels for the promotion of road safety ownership and accountability? | 94 | 93 |
| What are the economic benefits of the installation of safety features during road construction, regular maintenance, and upgrading of roads? | 82 | 93 |
| How should different types of roads, and roads in different geographical locations, be designed to make them safer for all road users? | 82 | 93 |
| **Pillar 3: Safer vehicles** | | |
| What are the factors affecting fitness condition and roadworthiness of vehicles to the extent that it leads to road traffic crashes? | 86 | 100 |
| What should be the minimum criteria for the establishment of standard vehicular maintenance workshops? | 93 | 92 |
| What are the capacity development and training needs for currently working human resources and additional jobs to improve the safety of vehicles in Nepal? | 71 | 92 |
| What improvements in policies and institutional setup are needed to ensure vehicle safety of all types and routes? | 79 | 92 |
| What is the role of motor parts used for vehicle maintenance for fitness condition of the vehicles and road crashes? | 93 | 83 |
| How does overloading impact the safety of the vehicles? | 71 | 83 |
| What are the vehicle-related factors causing road crashes in Nepal? | 71 | 83 |
| **Pillar 4: Safer road users** | | |
| How can the driver licensing system be made more effective to ensure safer vehicle drivers? | 100 | 93 |
| What are the main factors increasing the risk of public vehicle crashes? What interventions would improve the safety of travel on public vehicles? | 94 | 93 |
| How can licensing and crash data collection systems be improved? | 94 | 93 |
| What are the major causes of road crashes in Nepal? What percentage of road crashes are due to unsafe road user behaviours? | 94 | 87 |
| What content should be included in awareness campaigns for different types of road users, and how are these campaigns best delivered? | 83 | 87 |
| What are the barriers to the implementation of laws regarding safer road user behaviour? Review of existing policies related to safer road users. | 78 | 87 |
| **Pillar 5: Postcrash response** | | |
| What standards should be applied to ambulance services? (includes standards for personnel and training, equipment carried, and the vehicles) | 100 | 100 |
| What is the standard of care at health centres and hospitals for road traffic injury patients across the country, and how can they be improved? | 79 | 92 |
| What is the current average time taken for a road traffic injury patient to receive first response at the scene and the average time taken to arrive at a healthcare setting able to meet their care needs? How can any delays be reduced? | 93 | 92 |

**Table 3** Continued

| | Scores* | |
|---|---|---|
| What factors influence the ability of the postcrash emergency response service to get to the patient and then get them to the right hospital in the best possible time? | 86 | 92 |
| What should be included in the training curriculum for the different levels of postcrash responders? | 93 | 85 |
| How should policies and legislation be further developed to support the postcrash response for road traffic injury victims? | 71 | 85 |
| What is the optimal model of insurance to minimise death & disability following a road traffic crash? What are the barriers and facilitators to implementing such an insurance system? | 71 | 85 |

The phrasing of questions presented in this table reflects the direct translation from Nepali to English of the research questions used in the ranking process.
*Percent of participants ranked 'very important' or 'important'; R2=round 2; R3=round 3.

road crashes,[33] and developing a national road safety education programme.[34] We have not identified any previously published Delphi studies that have included all five pillars of road safety in a single study.

Zhu et al[26] recruited road safety experts and young drivers in the USA to study the risks of mobile phone use while driving. Expert participants identified texting, sending emails, or picking up the phone as particularly high-risk behaviours for crashes, but not playing music on a handheld mobile which was prioritised by young drivers. Participants identified 20 behavioural practices related to mobile phone use which can result in a collision. Our study participants in pillar 4 also identified the importance of studying causes of driver distraction but did not identify mobile phone use in particular.

Cafiso et al[28] engaged the managers of large public bus companies in Italy in a Delphi study to explore bus safety. Participants rated safety solutions for issues relating to driver behaviour, traffic conflicts and vehicle maintenance and technology. Our study participants also raised concerns about the safety of public transport users and the safety of public passenger vehicles and prioritised a study to investigate the factors contributing to public vehicle crashes. The technological solutions explored in the study by Cafisco (eg, technology to control when the bus can start, automatic door closing, etc) are not applicable in the context of Nepal where public passenger vehicles are older and poorly equipped. An expert panel on sleep deprivation in a study by Czeisler et al,[27] agreed that a driver was not fit to drive if they had less than 2 hours of sleep in the previous 24 hours. In our study, participants raised concerns regarding driver behaviour, including fatigue but prioritised a study to review the entire driver licensing system rather than focusing on tackling specific driver behaviours. These examples illustrate how previous Delphi studies have tended to focus on specific road safety issues, and how the results are specific to the context or participants. Neither of these studies would be directly generalisable to Nepal, nor do they cover the breadth of safety issues identified in our study.

Several Delphi studies have reported postcrash trauma management and prehospital care. In Vietnam, Schmucker et al[29] used online meetings followed by a questionnaire survey of 1000 road users to generate responses that were ranked, and outcomes were used to inform the development of a trauma care course. Our study participants for pillar 5 also prioritised the development of training curricula for different levels of postcrash trauma care (table 3). Recently, Azami-Aghdash et al[30] used the Delphi technique to achieve a consensus on 37 indicators to measure and improve the performance of prehospital care following road crashes in Iran. This is similar to the topic prioritised for postcrash response (pillar 5) in our study. However, the differences in Iranian

| **Table 4** | Top six most urgent research questions |
|---|---|
| **Pillars** | **Research questions** |
| Pillar 1 | How can implementing agencies be made more accountable for road safety in urban and rural areas? |
| Pillar 2 | How should different types of roads, and roads in different geographical locations, be designed to make them safer for all road users? |
| Pillar 3 | What are the factors affecting fitness condition and road worthiness of vehicles to the extent that it leads to road traffic crashes? |
| Pillar 4 | How can the driver licensing system be made more effective to ensure safer vehicle drivers? |
| | What are the main factors increasing the risk of public vehicle crashes? What interventions would improve the safety of travel on public vehicles? |
| Pillar 5 | What factors influence the ability of the postcrash emergency response service to get to the patient and then get them to the right hospital in the best possible time? |

and Nepali country contexts and prehospital care infrastructure mean that performance indicators in Iran are not generalisable to Nepal. Balikuddembe et al[32] used the Delphi technique to identify and prioritise factors that could prevent and support victims of RTIs in Kampala. They identified 23 factors across the entire Emergency Medical Service system that were similar to issues raised by participants in pillar 5 of our study.

In the course of our study, shifts in the opinions of participants were observed during rounds 2 and 3. Concerning the rankings completed in round 2, a high degree of consensus was observed and the process of creating a reduced list for round 3 was relatively straightforward. The Delphi method dictates that the results of a first-round be represented to participants in subsequent Rounds, giving participants the opportunity to reconsider their views in the light of the discussion, additional thought and/or the results obtained from other participants.[20 35] Cafiso et al[28] in their study, similarly reported that after the second round, the Delphi panellists' opinions were influenced by those of their colleagues. In our study, the changed ranks of the questions between round 2 and round 3 illustrate the value and influence of discussion between rounds in reaching a consensus. High numbers of research questions were rated 'important' or 'very important' in our study, illustrating that many participants recognised the need for road safety research in Nepal. Issues relating to improving the safety of road users traditionally considered vulnerable (eg, pedestrians, cyclists, drivers and passengers of powered two wheelers) were raised by participants in this study, however, during ranking, research questions that improved the safety of all road users were prioritised over questions relating to these specific groups.

The government of Nepal plans to enact a Road Safety Bill[36] that will include issues relating to planning, resourcing, implementation and evaluation of national road safety activities. Provincial governments, which were established only 4 years ago, through the promulgation of the constitution of Nepal,[37] have started to enact Provincial Transport Management Acts. However, the institutional structures necessary to implement these laws are still in development.[13] The research questions prioritised in this study emphasise the need for evidence to support both national development plans[8] and safer roads and transport in Nepal.[38] Existing road safety policies are mostly only partially implemented.[13] Policy gaps include policies to separate traffic and road users and those to address speed management.

## Strengths and limitations

The high response rate (70%), and good representation and involvement of individuals and experts currently active in the fields of road construction, vehicle management, transport management and postcrash response is a major strength of this study. The Delphi method for achieving consensus is a research technique with the potential for biases[20]; Hallowell[17] outlined common biases in implementation and here we describe the measures applied to minimise these biases in this study. To minimise factors that might influence the quality of the conclusions due to the level of expertise of the panel members,[39] only experienced and recognised authorities working for road safety in Nepal were invited to participate. While most participants had a remit for national road safety, we acknowledge that 83/93 (89%) were from organisations based in Kathmandu valley which may have introduced a bias towards urban and highway crashes in the prioritised research questions. The results produced by Delphi studies may be considered limited due to the poor quality of the facilitator's survey instruments,[16] therefore, the tools developed for this study were informed by the international literature and advice was available from an experienced Delphi expert. Bias can occur if questions are poorly worded,[17] therefore, our researchers were trained in interviewing skills before commencing round 1 and conducted the interview in Nepali. Some critics believe that convergence of opinion in Delphi studies is conformity.[18] To counter this risk, we synthesised best global road safety practice as reported in published literature and presented this to participants during the workshops between rounds 2 and 3. This meant that participants ranked questions initially individually and then were allowed to change their minds after the group discussion. Although the Delphi approach has been reported to be time-intensive,[40] we found that the time taken to participate in this study did not significantly affect recruitment or retention. We successfully retained participants, as demonstrated by the fact that 64/93 (69%) participants were retained to round 3.

## CONCLUSIONS

This study identified research priorities for road safety in Nepal across all of the WHO's five pillars of road safety. The most urgent and important research questions related to: improving the governance of road safety through greater accountability, improving road design across different topographies, establishing the contribution of poor vehicle fitness to crash occurrence, strengthening the driver licensing system, improving the safety of passengers on public buses and understanding the barriers to the provision of effective postcrash care. These findings can guide researchers when designing future studies. In addition, the study provided opportunities for participants to meet stakeholders outside their sector and discuss the challenges identified. Future research has the potential to lead to evidence-informed policy development and implementation, and improve practices relating to road construction and management, vehicle standards and postcrash care, making the roads safer for all road users in Nepal.

**Acknowledgements** We would like to acknowledge the support of Professor Nichola Rumsey who provided training in Delphi study methodology to the study team, advised on the study protocol and this manuscript. We are grateful to all 93 expert participants without whom this study would not have been possible.

**Contributors** PRP, JM: design and administration of the study; data curation and analysis, original draft and finalisation. PR and KP: data collection; investigation; project administration; validation; review & editing. SKJ: project administration; resources; supervision; review & editing. JM: funding acquisition; methodology;

conceptualization; data curation; supervision; review & editing. All authors contributed to drafts and approved the final manuscript.

**Funding** This research was commissioned by the National Institute for Health Research (NIHR) Global Health Policy and Systems Research Development Award using UK aid from the UK Government (NIHR129877).

**Competing interests** None declared.

**Patient and public involvement** Patients and/or the public were not involved in the design, or conduct, or reporting, or dissemination plans of this research.

**Patient consent for publication** Not applicable.

**Ethics approval** Ethical approval for conducting this study was obtained from the Kathmandu Medical College Institutional Review Committee (ref. 040620201) and the University of the West of England Bristol Faculty Research Ethics Committee (ref. HAS. 20.06.192).

**Provenance and peer review** Not commissioned; externally peer reviewed.

**Data availability statement** Data are available on reasonable request.

**ORCID iDs**
Puspa Raj Pant http://orcid.org/0000-0002-8827-0018
Sunil Kumar Joshi http://orcid.org/0000-0002-2704-5060
Julie Mytton http://orcid.org/0000-0002-0306-4750

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
