## [Reviewer comments · BMJ Open]

ARTICLE DETAILS

TITLE (PROVISIONAL)	Identifying research priorities for road safety in Nepal: a Delphi study
AUTHORS	Pant, Puspa Raj; Rana, Pranita; Pradhan, Kriti; Joshi, Sunil Kumar; Mytton, Julie

VERSION 1 – REVIEW

REVIEWER	Madeline MacKechnie UCSF, Orthopaedic surgery
REVIEW RETURNED	15-Dec-2021

GENERAL COMMENTS	The authors present the results of a Delphi study that identified six road safety research priorities in Nepal through consensus of an expert panel. The 93 individuals that participated in the study represented five professional groups that were relevant to each of the five WHO pillars of road safety. The Delphi methodology was employed to develop consensus using three rounds (interviews, Likert scale survey scoring, and a workshop). While the topic of this manuscript is particular to Nepal, it also has a broader interest and value to the road safety community. Results from this study have the potential to lead to policy-changes and can help inform research and policy priorities for Nepal. Additionally, the study methodology could be used as a model for other regions/countries to determine their most pressing research priorities in road safety. Specific critiques: 1. The data analysis and summary statistics performed to determine the top ranked research questions and survey results from a 5-point Likert scale were not clearly described. How were the survey questions and research questions ranked (mean, median, etc.)?2. 83 of the 93 participants were from the Kathmandu valley, with only 10 from outside the area. Does this mean that most study participants were coming from a relatively urban setting? Are the results generalizable to all settings or do they likely only reflect priorities of those in urban settings (and do not account for those in rural settings)? This may need to be listed as a limitation.3. Is the 9% of the female participants consistent with the make-up of females in the professional backgrounds listed (government organizations, clinicians, nurses, physiotherapists, road safety engineers, etc.) in Nepal or are they underrepresented in this study?4. The 70% response rate (93 of 133 invited) for this study is quite impressive and could be listed as a strength.5. Line 10 (pg 14) delete "of".6. There are a few minor grammatical errors throughout the paper that should be reviewed to ensure that it reads as though it were
---

	written by a native English speaker.
--	--------------------------------------

REVIEWER	Apostolos Ziakopoulos National Technical University of Athens
REVIEW RETURNED	21-Jan-2022

GENERAL COMMENTS	The paper deals with a very interesting subject dealing with prioritizing road safety research in Nepal by collecting expert opinions via a Delphi study. The paper is well written and applies the Delphi methodology appropriately. The results are very interesting and quite fruitful as a contribution from a less-researched country in the road safety aspect. However, there is a number of issues to be addressed before the paper is ready for publication. Specifically: - In the abstract, on page 1, lines 50-52, the five pillars of safety should be briefly outlined.- On page 1, line 19, the referred timespan is unclear. Is the 5-year period from July 2018 to the present time? The authors are asked to clarify.- Towards the end of the Introduction section, it is customary to provide an outline of the study.- The reviewer was surprised not to find any mention or provision for vulnerable road users (e.g. pedestrians, cyclists, powered-two-wheeler occupants, mopeds or similar) in the Delphi questions. Is there a reason for this omission (which might have happened by the road safety experts, not the authors themselves)? Please discuss.- There are many parts in the Discussion Section that elaborate on previous studies without comparison to present results. These studies should form an independent Literature Review Section before the Results section. Only studies that are contrasted with present results should remain in the Discussion Section.- The research priorities identified by the study (Table 4) should be summarized in 2-3 sentences in the Abstract and Conclusions.- Some yellow highlighted text remains on page 6.- On Figure 2, the axes labels should be enlarged.
---

VERSION 1 – AUTHOR RESPONSE

Comments of Reviewer 1.

We thank Reviewer 1 for their supportive feedback on our study and manuscript.

Comment 1: The data analysis and summary statistics performed to determine the top ranked research questions and survey results from a 5-point Likert scale were not clearly described. How were the survey questions and research questions ranked (mean, median, etc.)?

Authors' response: Thank you for highlighting that our text was not clear on this issue. Each participant ranked each question on the five-point Likert scale. Where 70% of participants ranked a question as 'important' or 'very important', then that question was prioritised for the next round.

Therefore, we used the frequency of responses (not means or medians) to determine consensus. We have amended the text in paragraph 2 of the Data collection section to improve clarity on this issue.

Comment 2: 83 of the 93 participants were from the Kathmandu valley, with only 10 from outside the area. Does this mean that most study participants were coming from a relatively urban setting? Are

the results generalizable to all settings or do they likely only reflect priorities of those in urban settings (and do not account for those in rural settings)? This may need to be listed as a limitation.

Authors' response: We acknowledge the risk highlighted by the reviewer, though do not think that the high proportion of participants who were resident in the Kathmandu valley will have significantly skewed the responses towards urban crash issues. This is because many professionals and organisations working on road safety, whilst resident in Kathmandu, have a remit for the whole country. Questions specifically related to road traffic crashes on rural roads were recorded from participants, but these were not prioritised in later rounds. We have added this as a potential limitation of the study, and revised the text in the first paragraph of the results section to:

“Out of 93 participants, 83 were from Kathmandu valley represented organisations having remit to work or influence road safety nationally, and 10 were from outside Kathmandu and added value to the study by bringing in local contexts.”

Comment 3. Is the 9% of the female participants consistent with the make-up of females in the professional backgrounds listed (government organizations, clinicians, nurses, physiotherapists, road safety engineers, etc.) in Nepal or are they underrepresented in this study?

Authors' response: Female participation in road safety is known to be low in Nepal and we were pleased that we managed to recruit 9/93 (10%) female participants. The Nepali Labour Force Survey in 2004 reported that nearly 80% of Nepali females “did not work outside home”, and as recently as 2012 the percentage of civil service positions filled by women was only 15%. The 2018 Labour Force Survey of Nepal found that only about 11% of the labour force engaged in the five “industries” viz. construction, trade and repair of motor vehicles, transportation, professional & technical activities and Human Health were women. Therefore, we do not believe that women were under-represented in our study. We have added a sentence to the first paragraph of the discussion to explain this.

Comment 4. The 70% response rate (93 of 133 invited) for this study is quite impressive and could be listed as a strength

Authors' response: Thank you for your suggestion. We have added this as a bullet point and also mentioned the text under the heading of “strengths and limitations”

Comment 5: Line 10 (pg 14) delete “of”

Authors' response: corrected

Comment 6: There are a few minor grammatical errors throughout the paper that should be reviewed to ensure that it reads as though it were written by a native English speaker

Authors' response: Thanks for pointing it out, the manuscript has been checked by the native English-speaking co-author and amendments made.

C. Comments of Reviewer 2.

Comment 1: In the abstract, on page 1, lines 50-52, the five pillars of safety should be briefly outlined.

Authors' response: Thank you for your suggestion. The abstract has been extensively revised and now includes the sentence: “covering all five of the World Health Organisation's ‘pillars’ of road safety: 1) Road Safety Management; 2) Safer Roads; 3) Safer Vehicles; 4) Safer Road Users and 5) effective post-crash response.”

Comment 2: On page 1, line 19, the referred timespan is unclear. Is the 5-year period from July 2018 to the present time? The authors are asked to clarify.

Authors' response: Thank you for highlighting this point. The text has been revised as: “between July 2013 and July 2018”

Comment 3: Towards the end of the Introduction section, it is customary to provide an outline of the study

Authors' response: Thank you for your suggestion. The last line of introduction now reads: “Therefore, this study aimed to – involve a wide range of experts and participants representing stakeholder organisations to identify the research needed to help agencies in Nepal develop a safe systems approach to road safety, and to achieve a consensus about which studies should be prioritised”

Comment 4: The reviewer was surprised not to find any mention or provision for vulnerable road users (e.g. pedestrians, cyclists, powered-two-wheeler occupants, mopeds or similar) in the Delphi

questions. Is there a reason for this omission (which might have happened by the road safety experts, not the authors themselves)? Please discuss.

Authors' response: We thank the reviewer for this important observation. Research issues for road users traditionally considered vulnerable were raised by participants at the interview stage, but interestingly, research questions relating to these groups were not prioritised at the ranking stage over research questions that aimed to improve the safety of all road users. We have added a sentence to the second last paragraph of the discussion section to explain this finding.

Comment 5: There are many parts in the Discussion Section that elaborate on previous studies without comparison to present results. These studies should form an independent Literature Review Section before the Results section. Only studies that are contrasted with present results should remain in the Discussion Section

Authors' response: We thank the reviewer for this helpful observation. Our intention in the discussion was to highlight studies that contrasted with our findings, but we acknowledge this was not clear in our original submission. We have now revised the text throughout the Discussion section to better contrast previously published research with the findings of our study.

Comment 6: The research priorities identified by the study (Table 4) should be summarized in 2-3 sentences in the Abstract and Conclusions.

Authors' response: We thank the reviewer for this suggestion. We have included the 6 prioritised questions in the results section of the Abstract and have summarised the prioritised areas in the Conclusions section.

Comment 7: Some yellow highlighted text remains on page 6

Authors' response: Please accept our apologies for not removing this. The highlight has been corrected

Comment 8: On Figure 2, the axes labels should be enlarged

Authors' response: The axes labels have been revised and a new image uploaded into ManuscriptCentral

VERSION 2 – REVIEW

REVIEWER	Madeline MacKechnie UCSF, Orthopaedic surgery
REVIEW RETURNED	25-Feb-2022

GENERAL COMMENTS	Overall, the study is interesting and well-designed and the revisions have improved the paper. However, I have listed below minor grammatical edits/comments that the authors should address in order to strengthen the manuscript. Thank you. ABSTRACT Abstract revisions are better and clearer. Page 38 line 15 in Results – What do you mean “93 participants took part in a total of 95 interviews”? This is confusing. Do you mean that 93 participants were a part of the expert panel that underwent two rounds of interviews? If so, I'd clarify. Page 39 line 13 – Grammatical edit. The sentence should say “bringing perspectives...” Page 39 Line 21 – Suggest deleting the sentence “The Delphi approach is at risk of high dropout of participants” and instead just start with “We were able to retain...” INTRODUCTION Page 39 line 49 – Replace “huge” with “large”.
--

METHODS

Page 41 Line 22 – How were the authors able to know the denominator (133) and response rate (70%) if you used snowball method among multiple networks? In this case, it is usually very difficult to know who is receiving the survey. Did you ask each organization to report how many individuals the survey link was sent out to?

Page 42 line 21 – For consistency, the 5-point Likert scale of importance should be “Not important”, “Slightly important”, “Moderately important”, “Important”, “Very important”.

Page 42 line 51 – all mentions of “Covid-19” should be COVID-19. Also, the first time you mention it (I believe on line 51) the authors should introduce the disease (e.g. the novel coronavirus disease (COVID-19) pandemic...).

RESULTS

Page 41 line 40 – “Out of 133”. Of should not be capitalized. Mention “Out of a total of 133...”

DISCUSSION

Page 49 line 25 – Grammatical edit. Should say “reach” not “approached”.

Also, instead of writing “(93 people)”, it would be stronger if the authors wrote “and 70% of participants (n=93) agreed to take part.”

Page 49 line 33 – Instead of “known to be” I suggest just writing “is low”.

Page 49 lines 41 – Add “response” to the sentence. It should state “This response rate is higher...”

Page 49 lines 47-48 – Delete the final sentence “Compared to these rates, participation in our study was good”. Already been stated above.

Page 50 line 33 – Grammatic edit. Should say “sending emails” not “sending email”.

Page 51 lines 25 – Delete “when ranking”.

Page 52 line 12 – Delete “view”.

Page 53 line 13 – Delete “overly”.

Page 52 line 14 – Suggest re-wording this sentence to something along the lines of “we found that the time taken to participate in this study did not affect the high response rate”.

Page 53 line 20 – Grammatical edit. This sentence should say “questions related to: improving the governance” not “questions related to; improving...”

Page 53 line 26 – Grammatic edit. Suggest the authors delete “and the study provided” and instead write “and provide opportunities for stakeholders...”.

Page 53 line 27 – Rather than stating “stakeholders meeting to debate these issues”, I suggest the authors focus on the outcomes

	that can come from the study's findings. For example, focus on how stakeholders can meet to work towards implementing/designing future programs that can improve the issues identified in the study. Page 53 line 28 – Grammatical edit. This sentence should say “improve practices relating to...”
REVIEWER	Apostolos Ziakopoulos National Technical University of Athens
REVIEW RETURNED	03-Mar-2022
GENERAL COMMENTS	The authors have taken into account all of the previous comments and they have addressed them to a satisfactory degree. The paper appears ready for publication and should therefore be accepted.

VERSION 2 – AUTHOR RESPONSE

Comments of Reviewer: 1

Comment. Overall, the study is interesting and well-designed and the revisions have improved the paper. However, I have listed below minor grammatical edits/comments that the authors should address in order to strengthen the manuscript. Thank you.

Authors' response: *We thank the reviewer for their positive feedback and meticulous observations including grammatical edits and typo corrections.*

ABSTRACT

Comment. Page 38 line 15 in Results – What do you mean “93 participants took part in a total of 95 interviews”? This is confusing. Do you mean that 93 participants were a part of the expert panel that underwent two rounds of interviews? If so, I'd clarify.

Authors' response: *Thank you for the query, we have revised sentence to: “A total of 93 participants took part in interviews and two rounds of ranking”. Two of our participants had expertise in more than one pillar of road safety. Therefore, these two participants were each interviewed twice; once for each pillar where they held expert knowledge. We recognise that our original statement was potentially confusing and have therefore simplified this sentence.*

Comment. Page 39 line 13 – Grammatical edit. The sentence should say “bringing perspectives...”

Authors' response: *corrected*

Comment. Page 39 Line 21 – Suggest deleting the sentence “The Delphi approach is at risk of high dropout of participants” and instead just start with “We were able to retain...”

Authors' response: *corrected*

INTRODUCTION

Comment. Page 39 line 49 – Replace “huge” with “large”.

Authors' response: *corrected*

METHODS

Comment. Page 41 Line 22 – How were the authors able to know the denominator (133) and response rate (70%) if you used snowball method among multiple networks? In this case, it is usually very difficult to know who is receiving the survey. Did you ask each organization to report how many individuals the survey link was sent out to?

Authors' response: Thank you for your comment. We asked participants if they knew of other experts who could contribute to the Delphi study and then we invited those additional people. We did not ask participants to cascade information about the study amongst their contacts. Therefore, we were able to keep track of the number of people we invited to participate in the study. None of the participants completed a ranking exercise without having first taken part in an interview. We have added a line to the recruitment section of the methods to make it clear that potential new participants were notified to the research team who then approached them.

Comment. Page 42 line 21 – For consistency, the 5-point Likert scale of importance should be “Not important”, “Slightly important”, “Moderately important”, “Important”, “Very important”.

Authors' response: corrected

Comment. Page 42 line 51 – all mentions of “Covid-19” should be COVID-19. Also, the first time you mention it (I believe on line 51) the authors should introduce the disease (e.g. the novel coronavirus disease (COVID-19) pandemic...).

Authors' response: corrected

RESULTS

Comment. Page 41 line 40 – “Out of 133”. Of should not be capitalized. Mention “Out of a total of 133...”

Authors' response: corrected

DISCUSSION

Comment. Page 49 line 25 – Grammatical edit. Should say “reach” not “approached”. Also, instead of writing “(93 people)”, it would be stronger if the authors wrote “and 70% of participants (n=93) agreed to take part.”

Authors' response: Thank you for your suggestion. We have replaced the word ‘approached’ with “invited” and amended the rest of the sentence as suggested.

Comment. Page 49 line 33 – Instead of “known to be” I suggest just writing “is low”.

Authors' response: corrected

Comment. Page 49 lines 41 – Add “response” to the sentence. It should state “This response rate is higher...”

Authors' response: corrected

Comment. Page 49 lines 47-48 – Delete the final sentence “Compared to these rates, participation in our study was good”.

Authors' response: Deleted

Comment. Page 50 line 33 – Grammatic edit. Should say “sending emails” not “sending email”.

Authors' response: corrected

Comment. Page 51 lines 25 – Delete “when ranking”.

Authors' response: Deleted

Comment. Page 52 line 12 – Delete “view”.

Authors' response: Deleted

Comment. Page 53 line 13 – Delete “overly”.

Authors' response: Deleted

Comment. Page 52 line 14 – Suggest re-wording this sentence to something along the lines of “we found that the time taken to participate in this study did not affect the high response rate”.

Authors' response: Thank you for this suggestion. We have revised the sentence as: “we found that the time taken to participate in this study did not significantly affect recruitment or retention.”

Comment. Page 53 line 20 – Grammatical edit. This sentence should say “questions related to: improving the governance” not “questions related to; improving...”

Authors' response: corrected

Comment. Page 53 line 26 – Grammatic edit. Suggest the authors delete “and the study provided” and instead write “and provide opportunities for stakeholders...”.

Authors' response: corrected

Comment. Page 53 line 27 – Rather than stating “stakeholders meeting to debate these issues”, I suggest the authors focus on the outcomes that can come from the study’s findings. For example, focus on how stakeholders can meet to work towards implementing/designing future programs that can improve the issues identified in the study.

Authors' response: Thank you for your comment. We have revised the sentence to read; “These findings can guide researchers when designing future studies. In addition, the study provided opportunities for participants to meet stakeholders outside their sector and discuss the challenges identified”.

Comment. Page 53 line 28 – Grammatical edit. This sentence should say “improve practices relating to...”

Authors' response: corrected

C. Reviewer: 2

Comments to the Author: The authors have taken into account all of the previous comments and they have addressed them to a satisfactory degree. The paper appears ready for publication and should therefore be accepted.

Authors' response: Thank you very much.